# OpenReview forum: "Test-Time Scaling in Reasoning Models Is Not Effective for Knowledge-Intensive Tasks Yet"
_ICLR.cc/2026/Conference — Submitted to ICLR 2026_

### Official Review · Reviewer_ESux · 2025-10-28

**Soundness:** 3
**Presentation:** 3
**Contribution:** 3
**Rating:** 6
**Confidence:** 4

**Summary:**

This paper investigates whether test-time scaling is effective for knowledge-intensive tasks that require high factual accuracy and low hallucination, similar to the observations for reasoning-intensive tasks such as mathematical reasoning. Though enabling thinking performs better than non-thinking LLMs, increasing test-time scaling cannot result in consistent improvements in factual accuracy and often fails to reduce hallucinations based on experiments on reasoning models. Note that the "test-time scaling" for open-sourced models such as DeepSeek-R1-Distill models actually refers to appending "wait" to extend the model's thinking and for proprietary models the thinking tokens/time is increased in the API call configurations.

**Strengths:**

+ The experiments are comprehensive, conducted using 14 models on two benchmarks.
+ The observations may be inspiring for future work in this direction about how to improve reasoning models on knowledge-intensive tasks.
+ The authors conducted a fine-grained analysis by comparing the performance between low-reasoning and high-reasoning modes and identified the correlation between the models' willingness to answer and hallucination changes, which may to some extent reflect the incapability of LLMs in answering specific questions.
+ several case studies are conducted on some of the most powerful proprietary models.

**Weaknesses:**

+ The evaluation is based on one LLM (GPT-4o-mini). As it is easy for LLMs to be hacked even when reference answers are provided, the authors should sample some test cases and report human agreement. The authors could also evaluate the results using two LLM evaluators.
+ No promising solutions are proposed. The authors are encouraged to alleviate this problem by attempts such as using a confidence threshold to force abstention when confidence is low to reduce hallucination at a reasonable computational cost.

**Questions:**

see the weakness

---

> ### Author Response · Authors · 2025-11-23
> **Reply by Authors**
>
> Thank you for your insightful feedback.
>
> > The authors should sample some test cases and report human agreement.
>
> We randomly sampled 300 cases and compared the automatic evaluation results with human annotations. We found only 2 disagreements, indicating that gpt-4o-mini serves as a reliable evaluator.
>
> > The authors could also evaluate the results using two LLM evaluators.
>
> We evaluated the responses using the latest frontier model gpt-5.1-2025-11-13. As shown in Table 1, the results are highly consistent with those from gpt-4o-mini. The agreement rate between the two evaluators is higher than 98% across all cases, further validating our automatic evaluation.
>
> Table 1: Experimental results evaluated by gpt-4o-mini and gpt-5.1-2025-11-13, and their agreement rates
>
> | Models             | Accuracy (%) evaluated by gpt-4o-mini | Hallucination ratio (%) evaluated by gpt-4o-mini | Accuracy (%) evaluated by gpt-5.1 | Hallucination ratio (%) evaluated by gpt-5.1 | Agreement rates between gpt-4o-mini and gpt-5.1 |
> | ------------------ | ------------------------------------- | ------------------------------------------------ | --------------------------------- | -------------------------------------------- | ----------------------------------------------- |
> | GPT-5-mini         | 22.8                                  | 47.9                                             | 22.6                              | 48.0                                         | 99.6                                            |
> | gpt-oss-20b        | 5.0                                   | 58.5                                             | 4.8                               | 59.0                                         | 99.3                                            |
> | Gemini 2.5 Flash   | 28.5                                  | 58.6                                             | 29.5                              | 57.9                                         | 98.4                                            |
> | Claude Sonnet 4    | 25.9                                  | 46.3                                             | 26.4                              | 45.8                                         | 98.5                                            |
> | Qwen3-8B           | 3.3                                   | 30.0                                             | 3.3                               | 30.1                                         | 99.6                                            |
> | R1-Distill-Qwen-7B | 0.9                                   | 71.5                                             | 0.9                               | 72.0                                         | 99.0                                            |
>
> > The authors are encouraged to alleviate this problem by attempts such as using a confidence threshold to force abstention when confidence is low to reduce hallucination at a reasonable computational cost.
>
> We explored a confidence-based abstention strategy and found that setting a confidence threshold can reduce hallucinations, but at the cost of lower accuracy, reflecting a precision-recall trade-off. We believe that mitigating this issue requires not only inference-time strategies, but also changes to training objectives and evaluation metrics. As discussed in Section 4.2, current test-time scaling methods often encourage models to produce an answer rather than express uncertainty as thinking time increases. This behavior may stem from the training and evaluation procedures, which rewards guessing over acknowledging uncertainty in order to maximize accuracy [1]. To mitigate this, future work should encourage models to appropriately express uncertainty rather than generating overconfident hallucinations when unsure.
>
> [1] Kalai, Adam Tauman, Ofir Nachum, Santosh S. Vempala, and Edwin Zhang. Why language models hallucinate. *arXiv preprint arXiv:2509.04664*

---

### Official Review · Reviewer_hS4G · 2025-10-31

**Soundness:** 1
**Presentation:** 2
**Contribution:** 1
**Rating:** 2
**Confidence:** 4

**Summary:**

This paper conducts case studies to understand the effect of test time scaling and more thinking on hallucination and factual accuracy of knowledge intensive tasks. They observe it may not reduce hallucination or increase accuracy, in fact in some cases increases it. Sometimes hallucinations may reduce but that often can be attributed due to model obstaining from further thinking.

**Strengths:**

There are some interesting observations from the case studies in the paper

**Weaknesses:**

- I feel the paper is better suited for a blog (or at best a short paper). I don’t think it has enough content or enough depth for a long paper
    - While the observations are interesting, they are not entirely new and some of them are indeed intuitive.  The paper simply does not go deep enough into each of these observations or prescribe any specific method to mitigate this.
    -  The conclusions are quite high level, many of them are just some observations from some instances or examples. While I am not judging the correctness of it, these are probably correct - just that I don’t see any fundamental contribution of the work (like some principled approach or some significant novelty) beyond just listing these observations.
    -
- Test time scaling is a very broad term, there are multiple ways to do that each driven by different signals and each having its own complexity. There are also more strategic (for e..g self-evolutionary or tree search ) mechanisms of iteratively improving something, which can also test time scaling methods.  In my understanding this paper has taken a very simple approach to test time scaling which is simply setting the thinking budget or effort parameters for specific frontier models. While very simple test time scaling strategies (like controlling budget parameters) may not effectively improve performance (or even degrade it), there are other more principled methods (MCTS, AlphaEvolve etc) which may not suffer from the same conclusions. For meaningfully doing such a study it has to be more comprehensive in terms of models and baselines used. For closed methods, undoubtedly there would be less control on the budget parameters. To get deeper insights it might be more interesting to examine open source models more closely.

**Questions:**

See the weakness section

---

> ### Author Response · Authors · 2025-11-24
> **Reply by Authors (1/)**
>
> We appreciate your insightful comments.
>
> > I don’t think it has enough content or enough depth for a long paper.
>
> > While I am not judging the correctness of it, these are probably correct. I don’t see any fundamental contribution of the work (like some principled approach or some significant novelty) beyond just listing these observations.
>
> We respectfully disagree with the assessment. Our work presents a comprehensive investigation of test-time scaling for knowledge-intensive tasks. We present **5** key findings:
> 1. Test-time scaling does not consistently improve accuracy.
> 2. Test-time scaling does not reduce hallucinations for most models. In some cases, extended reasoning even increases hallucinations.
> 3. Hallucination changes are largely driven by the model’s willingness to answer. Hallucination reductions often result from the model choosing to abstain after thinking more, while increased hallucinations come from the model attempting previously unanswered questions with longer reasoning.
> 4. Extended reasoning can induce confirmation bias, where models reinforce incorrect belief with fabricated details, resulting in overconfident hallucinations.
> 5. Compared to non-thinking, thinking is helpful for knowledge-intensive tasks. It improves accuracy and reduces hallucinations for most models.
>
> To support these findings, we conducted experiments on 14 reasoning models across two benchmarks, and studied three ways of test-time scaling strategies. We added additional experiments and analyses in the revised paper, including quantitative analysis on gpt-oss-20b and experiments with parallel sampling. These contributions go beyond observations, offering novel insights into why current test-time scaling strategies fail and highlighting important directions for future research. We kindly ask for a reassessment of the score of soundness, presentation, and contribution.
>
> > While the observations are interesting, they are not entirely new and some of them are indeed intuitive.
>
> To the best of our knowledge, our work is the first study to examine how increased test-time computation affects the performance of reasoning models on knowledge-intensive tasks. Our findings, such as thinking more can encourage models to attempt more questions, leading to more hallucinations, and extended reasoning induces confirmation bias, are entirely new. We did not see any prior work that reports the same findings.

---

> > ### Author Response · Authors · 2025-11-24
> > **Reply by Authors (2/)**
> >
> > > The conclusions are quite high level, many of them are just some observations from some instances or examples.
> >
> > >  To get deeper insights it might be more interesting to examine open source models more closely.
> >
> > We compare model responses at different reasoning levels and compute the ratio of *not attempted* and *correct*, to understand why thinking more affects hallucination outcomes in Section 4.2. Our results reveal that hallucination changes are driven by the model’s willingness to answer.
> >
> > In addition, as suggested by Reviewer KkBX, we conducted a quantitative analysis on the open-source model, gpt-oss-20b. We follow the approach in [1] to analyze when the model starts hallucinating during the reasoning process. Specifically, for cases where the model produces an incorrect final answer, we locate the token position where the final answer first appears in the reasoning trace. We then normalize this position by the total number of reasoning tokens. A lower value indicates the hallucination occurred earlier in the reasoning .
> >
> > Experiment results (Table 1) show that as test-time computation increases, the hallucinated answer tends to appear earlier. We then visualize all occurrences of the incorrect final answer in the reasoning traces (Figure 5 in the revised paper). The visualization shows that these incorrect answers reoccur many times, and the frequency of such recurrences increases with longer thinking. These recurrences are often accompanied by fabricated details. This suggests that extended reasoning can lead to earlier and more frequent factual errors. This further supports our conclusion that thinking more leads to confirmation bias, where models fabricate details to support initial belief. We have incorporated this analysis into Section 4.3 of the revised paper.
> >
> > Table1: Normalized token position of first appearance of incorrect final answers in the reasoning traces of gpt-oss-20b
> >
> > | Model              | Avg. position of first appearance of final answers on SimpleQA (%) | Avg. position of first appearance of final answers on FRAMES (%) |
> > | ------------------ | ------------------------------------------------------------ | ------------------------------------------------------------ |
> > | gpt-oss-20b-low    | 45.7%                                                        | 68.2%                                                        |
> > | gpt-oss-20b-medium | 29.9%                                                        | 45.4%                                                        |
> > | gpt-oss-20b-high   | 17.2%                                                        | 26.3%                                                        |
> >
> > [1] Chen, Xingyu, et al. "Do NOT Think That Much for 2+ 3=? On the Overthinking of Long Reasoning Models." Forty-second International Conference on Machine Learning.
> >
> > >The paper simply does not go deep enough into each of these observations or prescribe any specific method to mitigate this
> >
> > Our analysis in Section 4 shows that thinking more encourages models to attempt more questions, many of which become hallucinations. Moreover, extended reasoning can induce confirmation bias, where models reinforce incorrect beliefs with fabricated details.
> >
> > In the revised paper (Section 6), we discuss the limitation of current test-time scaling methods and propose guidelines to mitigate them. First, thinking more often encourages models to attempt more questions. This likely stems from current training and evaluation procedures, which favors guessing over acknowledging uncertainty in order to maximize accuracy [2].
> >
> > Second, current reasoning models are specifically trained on domains like coding or mathematics, where longer reasoning often helps by enabling diverse problem-solving and self-reflection. However, knowledge-intensive tasks primarily rely on factual recall, and verifying facts without external sources is difficult. As a result, models may struggle to manage extended reasoning, generating longer responses filled with fabricated details (Figure 5 in the paper). Addressing these limitations requires future work on training objectives and evaluation metrics that promote appropriate uncertainty expression and discourage fabrication when the model is unsure.
> >
> > [2] Kalai, Adam Tauman, Ofir Nachum, Santosh S. Vempala, and Edwin Zhang. Why language models hallucinate. arXiv preprint arXiv:2509.04664

---

> > > ### Author Response · Authors · 2025-11-24
> > > **Reply by Authors (3/)**
> > >
> > > >  In my understanding this paper has taken a very simple approach to test time scaling which is simply setting the thinking budget or effort parameters for specific frontier models.
> > >
> > > Test-time scaling methods we used have been shown to effectively improve model performance across various domains:
> > >
> > > 1. Increasing reasoning effort improves performance in domains such as visual reasoning, agentic coding, and graduate-level scientific problem solving [1]. This approach is officially recommended in model documentation [2].
> > > 2. Increasing thinking budget improves performance on benchmarks such as AIME 2025, LiveCodeBench and GPQA diamond (Figure 4 in [3]). It is also recommended in model documentation [4][5].
> > > 3. Extend model thinking with budget forcing improves performance in tasks such as MATH500, AIME 2025 and GPQA [6].
> > >
> > > [1] Introducing GPT-5: https://openai.com/index/introducing-gpt-5/
> > >
> > > [2] OpenAI documentation: Reasoning models. https://platform.openai.com/docs/guides/reasoning/advice-on-prompting
> > >
> > > [3] Comanici, Gheorghe, et al. "Gemini 2.5: Pushing the frontier with advanced reasoning, multimodality, long context, and next generation agentic capabilities." arXiv preprint arXiv:2507.06261 (2025).
> > >
> > > [4] Anthropic documentation: Building with extended thinking. https://platform.claude.com/docs/en/build-with-claude/extended-thinking
> > >
> > > [5] Google Gemini documentation: Gemini thinking. https://ai.google.dev/gemini-api/docs/thinking
> > >
> > > [6] Muennighoff N, Yang Z, Shi W, et al. s1: Simple test-time scaling. Proceedings of the 2025 Conference on Empirical Methods in Natural Language Processing. 2025: 20286-20332.
> > >
> > >
> > > > Test time scaling is a very broad term, there are multiple ways to do that each driven by different signals and each having its own complexity. There are also more strategic (for e..g self-evolutionary or tree search ) mechanisms of iteratively improving something, which can also test time scaling methods.
> > >
> > > > While very simple test time scaling strategies (like controlling budget parameters) may not effectively improve performance (or even degrade it), there are other more principled methods (MCTS, AlphaEvolve etc) which may not suffer from the same conclusions.
> > >
> > > We chose the test-time scaling methods used in our study for the following reasons:
> > >
> > > First, these strategies have demonstrated effectiveness across many domains and are recommended by model documentation, as discussed above.
> > >
> > > Second, other strategies, such as self-evolutionary or AlphaEvolve, typically require machine-gradable metrics to iteratively refine solutions. For example, AlphaEvolve[7] is proposed for coding tasks, where it can verify, run and score generated programs using automated evaluation metrics, and then improve the programs. Such methods can be applied to domains where progress can be clearly and systematically measured. However, knowledge-intensive primarily rely on factual recall rather than derivation. And automatic evaluation without external sources is challenging. So the nature of the task constrains which test-time scaling methods can be applied.
> > >
> > > [7] Novikov, Alexander, et al. "AlphaEvolve: A coding agent for scientific and algorithmic discovery." arXiv preprint arXiv:2506.13131 (2025).

---

> > > > ### Author Response · Authors · 2025-11-24
> > > > **Reply by Authors (4/4)**
> > > >
> > > > > For meaningfully doing such a study it has to be more comprehensive in terms of models and baselines used.
> > > >
> > > > We conducted experiments on test-time scaling with a parallel approach (self-consistency) [1]. We sample 1, 5, 10, and 15 paths and aggregate the answers with majority vote (SC@1 to SC@15). Results in Table 2 support our conclusion that increasing test-time computation does not consistently improve accuracy. As reasoning tokens increase, accuracy remains almost unchanged for most models.
> > > >
> > > > Table 2: Accuracy and hallucination ratio with self-consistency on SimpleQA
> > > >
> > > > | Models and self-consistency (SC) settings | Avg. reasoning tokens | Accuracy (%) | Hallucination Ratio (%) |
> > > > | ----------------------------------------- | --------------------- | ------------ | ----------------------- |
> > > > | GPT-5-mini SC@1                           | 295.8                 | 22.1         | 40.6                    |
> > > > | GPT-5-mini SC@5                           | 1446.0                | 23.3         | 38.5                    |
> > > > | GPT-5-mini SC@10                          | 2905.7                | 23.9         | 34.9                    |
> > > > | GPT-5-mini SC@15                          | 4357.1                | 23.8         | 32.8                    |
> > > > | O3-mini SC@1                              | 370.0                 | 8.9          | 14.4                    |
> > > > | O3-mini SC@5                              | 1798.9                | 9.9          | 11.9                    |
> > > > | O3-mini SC@10                             | 3599.0                | 9.6          | 11.1                    |
> > > > | O3-mini SC@15                             | 5397.1                | 9.3          | 10.4                    |
> > > > | Gemini-2.5-Flash SC@1                     | 439.2                 | 28.5         | 58.6                    |
> > > > | Gemini-2.5-Flash SC@5                     | 2083.3                | 30.1         | 49.4                    |
> > > > | Gemini-2.5-Flash SC@10                    | 4151.3                | 31.9         | 45.9                    |
> > > > | Gemini-2.5-Flash SC@15                    | 6226.1                | 32.8         | 43.8                    |
> > > > | Qwen3-8B SC@1                             | 617.8                 | 3.3          | 30.0                    |
> > > > | Qwen3-8B SC@5                             | 3044.2                | 3.6          | 16.5                    |
> > > > | Qwen3-8B SC@10                            | 6100.8                | 3.0          | 11.8                    |
> > > > | Qwen3-8B SC@15                            | 9145.4                | 3.0          | 9.4                     |
> > > > | R1-Distill-Qwen-14B SC@1                  | 527.9                 | 4.1          | 81.1                    |
> > > > | R1-Distill-Qwen-14B SC@5                  | 2705.3                | 5.1          | 70.1                    |
> > > > | R1-Distill-Qwen-14B SC@10                 | 5409.1                | 5.5          | 60.1                    |
> > > > | R1-Distill-Qwen-14B SC@15                 | 8114.8                | 5.4          | 56.5                    |
> > > >
> > > > We also observe that the hallucination ratio decreases with more samples. To understand this, we follow the method described in Section 4.1. Specifically, for questions where the hallucination outcome differs between SC@1 and SC@15 (i.e., SC@1 is hallucinated but SC@15 is not), we examine the non-hallucinating responses and compute the ratio of “correct” versus “not attempted”. As shown in Table 3, parallel sampling reduces hallucinations largely because it leads the model to abstain from answering, rather than improved factual accuracy. This supports our conclusion that hallucination changes are driven by the model’s willingness to answer. We have added the experiments to Appendix B in the revised paper.
> > > >
> > > > Table 3: The ratio of “not attempted” and “correct” in the non-hallucinating responses of SC@15
> > > >
> > > > | Models              | Not attempted (%) | Correct (%) |
> > > > | ------------------- | ----------------- | ----------- |
> > > > | GPT-5-mini          | 73.9              | 26.1        |
> > > > | o3-mini             | 79.6              | 20.4        |
> > > > | Gemini-2.5-Flash    | 56.1              | 43.9        |
> > > > | Qwen3-8B            | 96.7              | 3.3         |
> > > > | R1-Distill-Qwen-14B | 92.6              | 7.4         |
> > > >
> > > > [1]Wang, Xuezhi, et al. "Self-consistency improves chain of thought reasoning in language models." ICLR 2023.

---

### Official Review · Reviewer_KkBX · 2025-10-31

**Soundness:** 4
**Presentation:** 4
**Contribution:** 3
**Rating:** 6
**Confidence:** 5

**Summary:**

This paper investigates the impact of extending models' reasoning length on their performance in factual question-answering tasks. Through extensive experiments on 14 models and 2 benchmarks, the authors show that simply increasing the reasoning length does not consistently improve factual accuracy. The paper further analyzes the changes in "no attempt" rates and hallucination rates for different models, finding that longer reasoning often leads to more hallucinations.

**Strengths:**

1. Insightful conclusion: The main finding that longer reasoning does not help models recall facts they do not know, but can instead induce hallucinations, is a significant and insightful contribution. This challenges the common assumption that more computation uniformly leads to better performance.

2. Thorough experiments and nuanced analysis: The study is comprehensive, covering 14 models with different scaling mechanisms. The authors perform a detailed analysis of how different models behave, and the qualitative case studies are well-chosen to explain why hallucination rates change. The paper's clear distinction between the benefits of enabling thinking (vs. non-thinking) and the ineffectiveness of extending thinking is a crucial and well-supported nuance.

**Weaknesses:**

1. Superficial definition of hallucination: The paper's primary metrics rely on classifying the final answer as "correct," "incorrect," or "not attempted". Hallucination is simply defined as an "incorrect" final answer. This approach fails to analyze the emergence of hallucinations within the reasoning process itself. The paper's most interesting qualitative finding is that models fabricate facts during long reasoning, yet this is not systematically or quantitatively measured. Besides, the " overconfident hallucinations" is similar to the "overthinking" described in [1], but [1] focus on the correctness of reasoning process as well as the final answer. The study would be much stronger if it analyzed the factual correctness of the reasoning steps, not just the final output.

[1] Chen X, Xu J, Liang T, et al. Do NOT Think That Much for 2+ 3=? On the Overthinking of Long Reasoning Models[C]//Forty-second International Conference on Machine Learning.

2. Problematic experimental methods and limited scope: For Qwen3 and DeepSeek-R1-Distill models, the "budget forcing" method (appending "Wait") is a highly artificial intervention. This unnatural method of forcing a model to continue reasoning may be the cause of the observed hallucinations, rather than the extended reasoning per se. Furthermore, the paper's claim is about "test-time scaling" in general, but its investigation is limited to only one form: sequential (long CoT) scaling. It explicitly omits other traditional TTS methods, such as parallel approaches like multiple sampling and self-consistency. This limitation narrows the scope and impact of the main conclusion.

**Questions:**

1. When extending the reasoning length, does the frequency of factual errors (hallucinations) within the reasoning process itself increase?

2. The paper shows that models fabricate facts during long reasoning. In the tests on the FRAMES dataset, how many errors are due to such fabricated facts, and how many are due to multi-hop reasoning errors (e.g., errors in logic or composition)?

3. In the comparison between "thinking" and "non-thinking" models, what is the typical reasoning length (in tokens) for the "thinking" mode? How does this length compare to the "extended" reasoning lengths that were found to cause more hallucinations?

4. Does your analysis suggest that an optimal reasoning length selection strategy exists?

---

> ### Author Response · Authors · 2025-11-23
> **Reply by Authors (1/)**
>
> We appreciate your constructive feedback.
>
> > This approach fails to analyze the emergence of hallucinations within the reasoning process itself. The paper's most interesting qualitative finding is that models fabricate facts during long reasoning, yet this is not systematically or quantitatively measured. Besides, the " overconfident hallucinations" is similar to the "overthinking" described in [1], but [1] focus on the correctness of reasoning process as well as the final answer. The study would be much stronger if it analyzed the factual correctness of the reasoning steps, not just the final output.
>
> > When extending the reasoning length, does the frequency of factual errors (hallucinations) within the reasoning process itself increase?
>
> We follow the approach in [1] to analyze when the model starts hallucinating during the reasoning process. Specifically, for cases where the model produces an incorrect final answer, we locate the token position where the final answer first appears in the reasoning trace. We then normalize this position by the total number of reasoning tokens. A lower value indicates the hallucination occurred earlier in the reasoning . Experiment results (Table 1) show that as test-time computation increases, the hallucinated answer tends to appear earlier. We then visualize all the positions of the incorrect answer in the reasoning traces (Figure 5 in the revised paper). These incorrect final answers reoccur many times, and the frequency of such recurrences increases with longer thinking. These recurrences are often accompanied by fabricated details. This suggests that extended reasoning can lead to earlier and more frequent factual errors. This further supports our conclusion that thinking more leads to confirmation bias, where the models fabricate details to support initial belief. We have incorporated this analysis into Section 4.3 of the revised paper.
>
> Table1: Normalized token position of first appearance of incorrect final answers in the reasoning traces of gpt-oss-20b
>
> | Model              | Avg. position of first appearance of final answers on SimpleQA (%) | Avg. position of first appearance of final answers on FRAMES (%) |
> | ------------------ | ------------------------------------------------------------ | ------------------------------------------------------------ |
> | gpt-oss-20b-low    | 45.7%                                                        | 68.2%                                                        |
> | gpt-oss-20b-medium | 29.9%                                                        | 45.4%                                                        |
> | gpt-oss-20b-high   | 17.2%                                                        | 26.3%                                                        |
>
> [1] Chen X, Xu J, Liang T, et al. Do NOT Think That Much for 2+ 3=? On the Overthinking of Long Reasoning Models[C]//Forty-second International Conference on Machine Learning
>
> >  For Qwen3 and DeepSeek-R1-Distill models, the "budget forcing" method (appending "Wait") is a highly artificial intervention.This unnatural method of forcing a model to continue reasoning may be the cause of the observed hallucinations, rather than the extended reasoning per se.
>
> We append “Wait” to encourage models to think more before producing an answer. Our experimental results show that budget forcing does not reduce hallucinations for most Qwen3 and DeepSeek-R1-Distill models, nor does it increase them. The hallucination ratio fluctuates across different settings. One exception is DeepSeek-R1-Distill-Qwen-14B, where hallucinations decrease due to some responses changing from incorrect to abstention after extended reasoning.

---

> > ### Author Response · Authors · 2025-11-23
> > **Reply by Authors (2/)**
> >
> > > It explicitly omits other traditional TTS methods, such as parallel approaches like multiple sampling and self-consistency.
> >
> > We conducted experiments using a parallel approach (self-consistency). We sample 1, 5, 10, and 15 paths and aggregate the answers with majority vote (SC@1 to SC@15). Results in Table 2 support our conclusion that increasing test-time computation does not consistently improve accuracy. As the reasoning tokens increase, the accuracy remains almost unchanged for most models.
> >
> > Table 2: Accuracy and hallucination ratio with parallel sampling on SimpleQA
> >
> > | Models and self-consistency (SC) settings | Avg. reasoning tokens | Accuracy (%) | Hallucination Ratio (%) |
> > | ----------------------------------------- | --------------------- | ------------ | ----------------------- |
> > | GPT-5-mini SC@1                           | 295.8                 | 22.1         | 40.6                    |
> > | GPT-5-mini SC@5                           | 1446.0                | 23.3         | 38.5                    |
> > | GPT-5-mini SC@10                          | 2905.7                | 23.9         | 34.9                    |
> > | GPT-5-mini SC@15                          | 4357.1                | 23.8         | 32.8                    |
> > | O3-mini SC@1                              | 370.0                 | 8.9          | 14.4                    |
> > | O3-mini SC@5                              | 1798.9                | 9.9          | 11.9                    |
> > | O3-mini SC@10                             | 3599.0                | 9.6          | 11.1                    |
> > | O3-mini SC@15                             | 5397.1                | 9.3          | 10.4                    |
> > | Gemini-2.5-Flash SC@1                     | 439.2                 | 28.5         | 58.6                    |
> > | Gemini-2.5-Flash SC@5                     | 2083.3                | 30.1         | 49.4                    |
> > | Gemini-2.5-Flash SC@10                    | 4151.3                | 31.9         | 45.9                    |
> > | Gemini-2.5-Flash SC@15                    | 6226.1                | 32.8         | 43.8                    |
> > | Qwen3-8B SC@1                             | 617.8                 | 3.3          | 30.0                    |
> > | Qwen3-8B SC@5                             | 3044.2                | 3.6          | 16.5                    |
> > | Qwen3-8B SC@10                            | 6100.8                | 3.0          | 11.8                    |
> > | Qwen3-8B SC@15                            | 9145.4                | 3.0          | 9.4                     |
> > | R1-Distill-Qwen-14B SC@1                    | 527.9                 | 4.1          | 81.1                    |
> > | R1-Distill-Qwen-14B SC@5                    | 2705.3                | 5.1          | 70.1                    |
> > | R1-Distill-Qwen-14B SC@10                   | 5409.1                | 5.5          | 60.1                    |
> > | R1-Distill-Qwen-14B SC@15                   | 8114.8                | 5.4          | 56.5                    |
> >
> > We also observe that the hallucination ratio decreases with more samples. To understand this, we follow the method described in Section 4.1. Specifically, for questions where the hallucination outcome differs between SC@1 and SC@15 (i.e., SC@1 is hallucinated but SC@15 is not), we examine the non-hallucinating responses and compute the ratio of “correct” versus “not attempted”. As shown in Table 3, parallel sampling reduces hallucinations largely because it leads the model to abstain from answering, rather than improved factual accuracy. This supports our conclusion that hallucination changes are driven by the model’s willingness to answer. We added the experiments to Appendix B in the revised paper.
> >
> > Table 3: The ratio of “not attempted” and “correct” in the non-hallucinating responses of SC@15
> >
> > |  Models   | Not attempted (%) | Correct (%) |
> > | ----------------- | ----------------- | ----------- |
> > | GPT-5-mini        | 73.9              | 26.1        |
> > | o3-mini           | 79.6              | 20.4        |
> > | Gemini-2.5-Flash  | 56.1              | 43.9        |
> > | Qwen3-8B          | 96.7              | 3.3         |
> > | R1-Distill-Qwen-14B | 92.6              | 7.4         |

---

> > > ### Author Response · Authors · 2025-11-23
> > > **Reply by Authors (3/3)**
> > >
> > > > The paper shows that models fabricate facts during long reasoning. In the tests on the FRAMES dataset, how many errors are due to such fabricated facts, and how many are due to multi-hop reasoning errors (e.g., errors in logic or composition)?
> > >
> > > Our work primarily focuses on the factual accuracy of reasoning models. While FRAMES involves multi-hop reasoning, its logical structure is generally straightforward for current reasoning models. To assess the source of errors, we manually examined 100 error cases of gpt-oss-20b. In the low reasoning effort, only 2 errors were due to multi-hop reasoning failures. For the medium effort setting, we did not identify any errors caused by reasoning logic. So most of the errors are due to factual errors, rather than logical failures.
> > >
> > > > In the comparison between "thinking" and "non-thinking" models, what is the typical reasoning length (in tokens) for the "thinking" mode?
> > >
> > > We have added the reasoning length to Table 2 and Table 4 in the revised paper, and updated the results of Gemini 2.5 Flash and Claude Sonnet 4 from a lower thinking budget. We provide the test-time scaling settings for the “thinking” mode in Appendix A.4. Specifically, for Table 2 and Table 4, we use “low” reasoning effort for GPT-5 and GPT-5-mini. For Gemini 2.5 Flash and Claude Sonnet 4, we set the thinking budget to 256 and 1024 tokens, respectively. For Qwen3 models, we use the natural outputs without extension.
> > >
> > > > How does this length compare to the "extended" reasoning lengths that were found to cause more hallucinations?
> > >
> > > For GPT-5 and GPT-5-mini, increasing reasoning effort from “low” to “medium” or “high” leads to more hallucinations. For Gemini 2.5 Flash, further increasing the thinking budget also leads to more hallucinations. For Claude Sonnet 4 and Qwen3 models, further increasing the reasoning length does not reduce hallucinations. Therefore, enabling thinking can be helpful, but further extending the reasoning length is not effective for knowledge-intensive tasks yet.
> > >
> > > > Does your analysis suggest that an optimal reasoning length selection strategy exists?
> > >
> > > Our analysis suggests that enabling a low level of reasoning, such as “low” reasoning effort or a modest thinking budget, is generally helpful for knowledge-intensive tasks. Further increasing reasoning length beyond this point offers diminishing returns and may even lead to more hallucinations. From an efficiency perspective, a lightweight “thinking” mode is sufficient for knowledge-intensive tasks.

---

> > > > ### Comment · Reviewer_KkBX · 2025-11-24
> > > >
> > > > Thank you for your detailed replies and your efforts for rebuttle. The authors provided additional experimental results and addressed my concerns. Most of my concerns have been resolved. I believe the revised version is acceptable. I have no further questions.

---

### Official Review · Reviewer_8DPJ · 2025-11-01

**Soundness:** 3
**Presentation:** 3
**Contribution:** 2
**Rating:** 4
**Confidence:** 4

**Summary:**

The paper benchmarks reasoning models' behaviors with test-time scaling on two knowledge-intensive benchmarks, showing that more reasoning does not consistently improve accuracy or reduce hallucination for almost all models. There are also a few exemptions where more thinking budgets do help the model, and the authors provide case studies. Specifically regarding hallucination, experiments show that the changes are due to the models' willingness to answer. However, for models that support both "thinking" and "non-thinking" modes, the thinking mode does improve performance.

**Strengths:**

The paper investigates on sufficient model families, concluding basically consistent behaviors while also providing case studies. The main message is clear and the paper is well presented overall.

**Weaknesses:**

* While the number of models is large enough, the number of datasets / benchmarks seems to be insufficient. Can you elaborate on more datasets to show the universality?
* The system prompt is fixed across the paper. Did you try out different prompts? Would all prompts lead to similar conclusions?
* The main conclusion is interesting yet intuitive. The authors fail to provide any theoretical justification. What is the fundamental difference between thinking mode and non-thinking mode since they former leads to better performance? Why further increasing thinking budget would not lead to better performance?

Overall, this is a benchmarking paper with straightforward conclusions. It would be better if the authors could provide more theoretical analysis or intuitions, as well as appropriate solutions to improve model performance when the thinking budget is increased. Otherwise, the contribution is somewhat limited.

**Questions:**

See weaknesses.

---

> ### Author Response · Authors · 2025-11-23
> **Reply by Authors (1/)**
>
> Thank you for your insightful comments.
>
> > While the number of models is large enough, the number of datasets / benchmarks seems to be insufficient. Can you elaborate on more datasets to show the universality?
>
> We conducted experiments on SimpleQA Verified[1], a benchmark with 1,000 diverse knowledge-intensive questions. The dataset improves upon SimpleQA by removing duplicate sources and re-balancing topic and answer-type distributions to ensure question diversity.
>
> Experiment results (Table 1) show similar trends as in our main experiments. In particular, increasing test-time computation does not reduce hallucinations for most models, and can even increase them for models like GPT-5-mini and Gemini-2.5-Flash. This supports the generality of our findings. We also note that high-quality and challenging knowledge-intensive benchmarks are scarce, and many existing datasets are already saturated with current reasoning models. We have added the experiments on SimpleQA Verified to the revised paper in Appendix C.3.
>
> Table 1: Accuracy and hallucination ratio with increased test-time computation on SimpleQA Verified
> | Model and TTS settings               | Avg. reasoning tokens | Accuracy (%) | Hallucination Ratio (%) |
> | ------------------------------------ | --------------------- | -------------------- | ----------------------- |
> | gpt5-mini-low                        | 294.0                 | 20.0                 | 47.4                    |
> | gpt5-mini-med                        | 918.5                 | 21.0                 | 51.8                    |
> | gpt5-mini-high                       | 3178.7                | 22.5                 | 64.0                    |
> | Gemini-2.5-Flash (256 token budget)  | 196.2                 | 20.2                 | 50.2                    |
> | Gemini-2.5-Flash (512 token budget)  | 346.0                 | 23.7                 | 56.1                    |
> | Gemini-2.5-Flash (1024 token budget) | 438.1                 | 26.6                 | 56.5                    |
> | Gemini-2.5-Flash (2048 token budget) | 500.9                 | 28.1                 | 57.7                    |
> | Qwen3-8B (no extension)              | 607.4                 | 2.4                  | 27.3                    |
> | Qwen3-8B (extended 2 times)          | 843.8                 | 2.5                  | 24.3                    |
> | Qwen3-8B (extended 4 times)          | 1037.3                | 2.7                  | 27.7                    |
> | Qwen3-8B (extended 8 times)          | 1373.8                | 2.2                  | 27.6                    |
> | DS-R1-Qwen-14B (no extension)        | 550.7                 | 5.6                  | 74.8                    |
> | DS-R1-Qwen-14B (extended 2 times)    | 768.7                 | 5.2                  | 73.2                    |
> | DS-R1-Qwen-14B (extended 4 times)    | 956.1                 | 4.5                  | 69.7                    |
> | DS-R1-Qwen-14B (extended 9 times)    | 1291.1                | 3.7                  | 67.5                    |
>
> [1] Haas, Lukas, et al. "SimpleQA Verified: A reliable factuality benchmark to measure parametric knowledge." arXiv preprint arXiv:2509.07968 (2025).

---

> > ### Author Response · Authors · 2025-11-23
> > **Reply by Authors (2/2)**
> >
> > > The system prompt is fixed across the paper. Did you try out different prompts? Would all prompts lead to similar conclusions?
> >
> > We conducted experiments with the following prompt, which does not explicitly instruct the model to abstain when uncertain: *Give me the answer to the following question. Put your answer on its own line after 'Answer:'*
> >
> > Results (Table 2) indicate similar conclusions with our main results that increasing test-time computation does not consistently increase accuracy and reduce hallucinations. We added the experiment results to the revised paper in Appendix C.2
> >
> > Table 2: Accuracy and hallucination ratio with increased test-time computation using an alternative prompt
> >
> > | Model and TTS settings                | Avg. reasoning tokens | Accuracy (%) | Hallucination Ratio (%) |
> > | ------------------------------------- | --------------------- | ------------ | ----------------------- |
> > | gpt-5-low                             | 493.3                 | 48.9         | 47.8                    |
> > | gpt-5-med                             | 1282.9                | 48.9         | 48.9                    |
> > | gpt-5-high                            | 2924.4                | 49.6         | 48.8                    |
> > | o3-mini-low                           | 751.7                 | 14.1         | 69.3                    |
> > | o3-mini-medium                        | 2023.5                | 15.3         | 75.0                    |
> > | o3-mini-high                          | 4843.9                | 16.4         | 75.9                    |
> > | Gemini-2.5-Flash (256 tokens budget)  | 186.6                 | 27.8         | 71.8                    |
> > | Gemini-2.5-Flash (512 tokens budget)  | 289.2                 | 29.4         | 70.3                    |
> > | Gemini-2.5-Flash (1024 tokens budget) | 329.9                 | 29.3         | 70.1                    |
> > | Gemini-2.5-Flash (2048 tokens budget) | 345.5                 | 28.4         | 71.0                    |
> > | Qwen3-8B (no extension)               | 797.1                 | 4.5          | 93.5                    |
> > | Qwen3-8B (extended 2 times)           | 1060.0                | 5.5          | 92.3                    |
> > | Qwen3-8B (extended 4 times)           | 1284.0                | 5.4          | 92.8                    |
> > | Qwen3-8B (extendded 8 times)          | 1686.3                | 3.6          | 93.6                    |
> >
> >
> > > What is the fundamental difference between thinking mode and non-thinking mode since the former leads to better performance?
> >
> > Prior work [1] provides theoretical insights into why Chain-of-Thought (“thinking mode”) is effective to improve the accuracy of large language models. This aligns with our experiment results that enabling thinking leads to better performance than the non-thinking mode.
> >
> > [1] Zhiyuan Li, Hong Liu, Denny Zhou, and Tengyu Ma. Chain of thought empowers transformers to solve inherently serial problems. ICLR 2024.
> >
> > > Why further increasing thinking budget would not lead to better performance?
> >
> > > It would be better if the authors could provide more theoretical analysis or intuitions, as well as appropriate solutions to improve model performance when the thinking budget is increased.
> >
> > Our experiments show that increasing the thinking budget encourages the model to attempt more questions, many of which become hallucinations (Section 4.2). One possible reason is that current training and evaluation procedures reward guessing over acknowledging uncertainty [2]. This pushes models to answer even when unsure to maximize the accuracy. Mitigating this requires changes to training objectives and evaluation metrics, shifting focus from only accuracy to also rewarding appropriate abstention.
> >
> > Another reason relates to the nature of knowledge-intensive tasks. Reasoning models are specifically trained on domains like coding or mathematical, where longer reasoning often helps. When increasing the thinking budget, models may explore different solutions to solve these tasks. However, knowledge-intensive tasks primarily rely on factual recall, and verifying facts without external sources is challenging. As a result, models may struggle to manage extended reasoning, generating longer thinking content filled with fabricated details rather than improving factual accuracy. As shown in Section 4.3, longer reasoning can induce confirmation bias, where models fabricate details to support early incorrect belief, which utilizes a large number of thinking tokens and ultimately worsens the performance.
> >
> > [2] Kalai, Adam Tauman, Ofir Nachum, Santosh S. Vempala, and Edwin Zhang. Why language models hallucinate. arXiv preprint arXiv:2509.04664

---

### Author Response · Authors · 2025-11-23
**Revisions in Paper**

We thank all reviewers for their insightful comments. We have revised the paper with the following revisions (Differences are highlighted in **blue**):

1. Added a qualitative and quantitative analysis of hallucination behavior of gpt-oss-20b in Section 4.3 (Reviewer KkBX, hS4G)
2. Added experiments using parallel sampling in Appendix B (Reviewer KkBX, hS4G)
3. Added experiments with an alternative prompt and experiments on SimpleQA Verified in Appendix C (Reviewer 8DPJ)
4. Added human agreement for automatic evaluation in Section 3.1 (Reviewer ESux)
5. Added discussion on the limits of current test-time scaling methods in Section 6 (Reviewer 8DPJ, ESux, hS4G)
6. Updated Table 2 (comparison between think vs. non-think) with results of Gemini 2.5 Flash and Claude Sonnet 4 using a lower thinking budget (Reviewer KkBX)
7. Updated GPT-5 results to use the same prompt as other models.

---

### Author Response · Authors · 2025-12-02
**Global Response by Authors**

Dear reviewers and AC,

We sincerely appreciate your time and effort in reviewing our paper. As highlighted by the reviewers, our study presents a comprehensive (KkBX, ESux), insightful (KkBX) and clearly written (8DPJ) analysis of test-time scaling for knowledge-intensive tasks. Our findings are interesting (hS4G), significant, well-supported (KkBX), and inspiring (ESux) for future work.

We thank all reviewers for their constructive feedback. In response, we have carefully revised the manuscript (with changes highlighted in  **blue**) to address the concerns and improve clarity and depth. The revisions include:

- **Added qualitative and quantitative analysis of hallucination emergence** using token position tracking in reasoning traces (**Section 4.3 and Figure 5**), suggested by Reviewer KkBX and hS4G.

- **Added experiments with parallel sampling (self-consistency)** to investigate the impact of increased test-time computation in a parallel approach (**Appendix B, Figure 6 and Table 3**), as suggested by Reviewer KkBX and hS4G.

- **Included additional experiments** with an alternative prompt (**Appendix C.2 and Figure 7**) and on the SimpleQA Verified benchmark (**Appendix C.3 and Figure 8**), as suggested by Reviewer 8DPJ.

- **Added human agreement results** for automatic evaluation to improve the reliability (**Section 3.1**), as requested by Reviewer ESux.

- **Expanded the discussion (Section 6)** on why current test-time scaling strategies do not work well for knowledge-intensive tasks and proposed directions for future research (Reviewers 8DPJ, ESux, hS4G).

- **Updated Table 2 with revised results** for Gemini 2.5 Flash and Claude Sonnet 4 using a lower thinking budget to better support our main findings (Reviewer KkBX).

- **Updated GPT-5 results (Section 3.2, 3.3 and Table 2)** to use the same prompt as other models, which explicitly asks the model to abstain when unsure, for consistency. The results still support our conclusions.

In our response to reviewer hS4G, we also reemphasized the novelty and contributions of our work. We hope these revisions and responses help clarify and strengthen the findings and overall messages of the paper, and inspire future research.

Sincerely,

Authors

---

### Meta-Review · Area_Chair_oAty · 2025-12-12

**Summary:**

I concur with Reviewer hS4G that this purely empirical study lacks the substantive insights expected of a top-tier conference paper. Specifically: (1) it makes no fundamental or technical contribution; (2) its experimental scope—in terms of both models and datasets—is too limited to support meaningful conclusions.

**Reviewer Concerns:**

While some minor experimental concerns have been addressed, the two major issues identified in the "Summary" section persist as unresolved.

**Reviewer Scores:**

Reviewer KkBX (6) may raise their score to 8. However, Reviewers 8DPJ (4) and hS4G (2) are likely to maintain their low scores, as their major concerns, which are the primary grounds for rejection, remain unaddressed. Reviewer ESux (6) raised only two concerns: one is related to a minor experimental issue, which has likely been addressed in the rebuttal, while the other concerns a lack of a newly proposed solution, which remains unresolved.

---

### Decision · Program_Chairs · 2026-01-26

Reject